# FIXED: Frustratingly Easy Domain Generalization with Mixup

Wang Lu[1], Jindong Wang[2],*, Han Yu[3], Lei Huang[2], Xiang Zhang[4], Yiqiang Chen[5], Xing Xie[2]

[1]Tsinghua University    [2]Microsoft    [3]Nanyang Technological University
[4] University of North Carolina    [5] University of Chinese Academy of Sciences
luw12thu@sina.com.cn, jindong.wang@microsoft.com

Domain generalization (DG) aims to learn a generalizable model from multiple training domains such that it can perform well on unseen target domains. A popular strategy is to augment training data to benefit generalization through methods such as Mixup [1]. While the vanilla Mixup can be directly applied, theoretical and empirical investigations uncover several shortcomings that limit its performance. Firstly, Mixup cannot effectively identify the domain and class information that can be used for learning invariant representations. Secondly, Mixup may introduce synthetic noisy data points via random interpolation, which lowers its discrimination capability. Based on the analysis, we propose a simple yet effective enhancement for Mixup-based DG, namely domain-invariant Feature mIXup (FIX). It learns domain-invariant representations for Mixup. To further enhance discrimination, we leverage existing techniques to enlarge margins among classes to further propose the domain-invariant Feature MIXup with Enhanced Discrimination (FIXED) approach. We present theoretical insights about guarantees on its effectiveness. Extensive experiments on seven public datasets across two modalities including image classification (Digits-DG, PACS, Office-Home) and time series (DSADS, PAMAP2, UCI-HAR, and USC-HAD) demonstrate that our approach significantly outperforms nine state-of-the-art related methods, beating the best performing baseline by 6.5% on average in terms of test accuracy. The code is available at `https://github.com/jindongwang/transferlearning/tree/master/code/deep/fixed`.

## 1. Introduction

In recent years, deep learning has demonstrated useful capabilities and potential in many application domains [2, 3]. However, the performance of deep neural nets often deteriorates significantly when deployed on test data that exhibit different distributions from the training data. This is widely recognized as the *domain shift* problem [4]. For instance, activity recognition models trained on the data from adults are likely to fail when being tested on children's activities, and the performance of natural image classification models tends to perform poorly when tested on artistic paintings.

A common technique to address the problem is domain adaptation (DA) [5–7]. It learns to maximize model performance on a given target domain with the help of labeled source domains by bridging the distribution gap. However, DA relies on target domains, which makes DA less applicable in real-world scenarios that demand good generalization performance on *unseen* distributions. Domain generalization (DG) [8–10] has attracted increasing attention in recent years. DG aims to learn a generalizable model that can perform well on unseen distributions after being trained on multiple source domains. Existing work can be categorized into three types: 1) learn domain-invariant representations [6, 11], 2) meta-learning [9, 12, 13], and 3) data augmentation-based DG [14, 15].

In this paper, we focus on data augmentation, specifically Mixup [1] which is a simple but effective approach. Mixup generates new samples via linear interpolations between any two pairs of data. It increases the quantity and diversity of training data to boost the generalization of deep nets [16]. Mixup can be used for domain generalization directly [17]. Recent works, such as FACT [18] and MixStyle [14], have adapted it in computer vision tasks with application-specific knowledge. Despite the success of Mixup, an important research question remains open: *Is there any versatile Mixup learning strategy for general domain generalization problems?*

---

*Corresponding author.

First Conference on Parsimony and Learning (CPAL 2024).

Our specific interest is to enhance the capability of Mixup for general domain generalization based on theoretical and empirical analysis of its current limitations. First, we notice that vanilla Mixup cannot discern domain information and class information, which can negatively affect its performance due to the entangled domain-class knowledge. Second, Mixup in DG can easily generate synthetic noisy data points when two classes are close to each other. This reduces the discrimination of the classifier. We propose the domain-invariant Feature MIXup with Enhanced Discrimination (*FIXED*) approach, to address these limitations of Mixup. It incorporates domain-invariant representation learning into Mixup, which enables the diverse data augmentation with useful information for the generalized model. Then, FIXED introduces a large margin to reduce synthetic noisy data points in the interpolation process. It is a simple yet effective approach.

Through theoretical analysis, we present insights on the design rationale and superiority of FIXED. Note that our FIXED is not limited to specific applications and can be applied to general classification tasks, in contrast to existing Mixup methods which are designed for computer vision tasks (e.g., [14, 18]). We have conducted extensive experiments on seven benchmarks across two modalities: 1) image classification (image data) and 2) sensor-based human activity recognition (time series data). The results demonstrate significant superiority of FIXED over nine state-of-the-art approaches, outperforming the best baseline by **6.5%** in terms of average test accuracy on the domain time series generalization task which is still in its infancy.

To summarize, our contributions are three-fold:

- Simple yet effective algorithm: For DG, we propose FIX to enhance the diversity of useful information and implement FIXED by introducing the large margin to reduce synthetic noisy data during Mixup. FIXED remains quite simple but effective.
- New theoretical insights: We offer theoretical insights from both the cover range and class distance perspectives to explain the rationale behind our algorithm.
- Good Performance: We conduct comprehensive experiments on seven benchmarks across two modalities: image classification (image) and sensor-based human activity recognition (time series). Experimental results demonstrate the superiority of FIXED, especially with **6.5%** improvements for domain generalization in time series which is still in infancy.

## 2. Preliminaries

We follow the definition in [10]. In domain generalization, we are given $M$ labeled source domains $\mathcal{S} = \{\mathcal{S}^i | i = 1, \cdots, M\}$ and $\mathcal{S}^i = \{(\mathbf{x}_j^i, y_j^i)\}_{j=1}^{n_i}$ denotes the $i^{th}$ domain, where $n_i$ denotes the number of data in $\mathcal{S}^i$. The joint distributions between each pair of domains are different and denoted as $\mathbb{P}_{XY}^i \neq \mathbb{P}_{XY}^j, 1 \leq i \neq j \leq M$. The goal of DG is to learn a robust and generalized predictive function $h : \mathcal{X} \to \mathcal{Y}$ from the $M$ training sources to achieve minimum prediction error on an unseen test domain $\mathcal{S}_{test}$ with an unknown joint distribution (i.e., $\min_h \mathbb{E}_{(\mathbf{x},y) \in \mathcal{S}_{test}}[\ell(h(\mathbf{x}), y)]$). $\mathbb{E}$ is the expectation and $\ell(\cdot, \cdot)$ is the loss function. All domains, including the source domains and the unseen target domains, have the same input and output spaces (i.e., $\mathcal{X}^1 = \mathcal{X}^2 = \cdots = \mathcal{X}^M = \mathcal{X}^T \in \mathbb{R}^m$). $\mathcal{X}$ is the $m$-dimensional instance space and $\mathcal{Y}^1 = \mathcal{Y}^2 = \cdots = \mathcal{Y}^M = \mathcal{Y}^T = \{1, 2, \cdots, K\}$. $\mathcal{Y}$ is the label space and $K$ is the number of classes.

### 2.1. Background

Data augmentation is a common technique to cope with DG. Among existing methods, Mixup [1] is a popular data augmentation approach and has shown good performance in many fields. It constructs synthetic training samples based on two random data points:

$$\lambda \sim Beta(\alpha, \alpha), \tilde{\mathbf{x}} = \lambda \mathbf{x}_i + (1 - \lambda)\mathbf{x}_j, \tilde{y} = \lambda y_i + (1 - \lambda)y_j, \tag{1}$$

where $Beta(\alpha, \alpha)$ is the Beta distribution and $\alpha \in (0, \infty)$ is a hyperparameter that controls the strength of interpolation between feature-target pairs, recovering the ERM principle as $\alpha \to 0$. Mixup extends the training distribution by incorporating the intuition that linear interpolations of feature vectors should lead to linear interpolations of the associated targets into the training set. As a powerful data augmentation technique, Mixup has played a vital role in enhancing sample diversity in domain generalization problems [14, 17, 18].

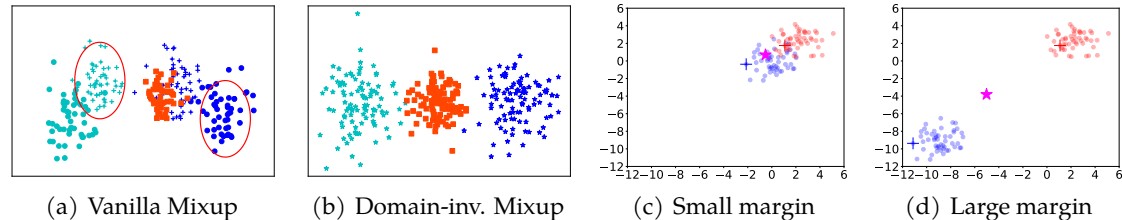

|  |  |  |  |
|---|---|---|---|
| (a) Vanilla Mixup | (b) Domain-inv. Mixup | (c) Small margin | (d) Large margin |

Figure 1: Toy examples to illustrate limitations of Mixup. Colors and shapes denote classes and domains, respectively. (a) Mixup generates unrecognizable synthetic data. (b) Mixup with only class information can mitigate such an issue. (c) Mixup tends to generate noisy data. (d) The large margin can reduce generations of synthetic noisy data.

## 2.2. Limitations of Mixup-based DG

Although the vanilla Mixup method can enhance data diversity, it fails to discern which features are useful for training the model. It only increases the diversity of all features equally. In DG, it cannot distinguish domain features and classification features, which results in the entanglement of domains and classes. It is unclear which parts of the increased diversity are useful for matching the categories. When incorrect matching between categories and features occurs, vanilla Mixup negatively affects model performance due to the introduction of interfering information. Figure 1(a) shows that vanilla Mixup directly mixes data without discerning classification and domain information. When mixing data points with the cyan "+"s and blue "o"s (circled in red), the red square data points are generated. Since the red square points are generated by two different classes, their labels should be in between the two classes, which means these points should lie between the two classes. However, as can be observed from Figure 1(a), the red squares almost completely overlap with the blue"+"s, which means they prefer to be the blue class according to the locations. Mixed domain information interferes with the matching of synthetic data points and synthetic labels. Not only vanilla Mixup, but also some adapted Mixup variants (e.g., manifold Mixup [19] which mixes in the hidden states) have the same limitation.

On the other hand, Mixup in DG is more likely to generate noisy synthetic data points [20]. Even when samples from different classes in the same domain are away from each other, data points from another domain with different distributions may be close to a cluster with a different category. When two clusters are close to each other, noisy synthetic data points are more likely to be generated. As shown in Figure 1(c), the blue cluster and the red cluster are very close. Noisy data points (e.g., the synthetic data points generated by the red "+"s and the blue "+"s) are generated with a high probability by Mixup.

## 3. The Proposed FIXED Method

In this paper, we propose the domain-invariant Feature MIXup with Enhanced Discrimination (FIXED) to address the aforementioned limitations of Mixup-based DG methods. The model architecture of FIXED is illustrated in Figure 2. We introduce its two critical modules as follows.

### 3.1. FIX: Domain-invariant Feature MIXup

We first introduce *domain-invariant* feature Mixup to discern the domain and class information, which is our main contribution. As suggested in [6], domain-invariant features contain more informative knowledge for classification than raw data [1] or the manifold Mixup [19]. Such feature Mixup is general and can be embedded in many existing DG methods. Let $\mathbf{z}$ be the domain-invariant feature. Then, our approach can be formulated as:

$$\lambda \sim Beta(\alpha, \alpha), \tilde{\mathbf{z}} = \lambda \mathbf{z}_i + (1 - \lambda)\mathbf{z}_j, \tilde{y} = \lambda y_i + (1 - \lambda)y_j. \tag{2}$$

Note that this is *not* the same as manifold Mixup [19] which operates on random layers and does not involve domain-invariant feature learning. Although domain-invariant feature learning alone brings about improvements for generalization, they usually lack diversity due to restrictions on the

learning process. Therefore, increasing diversity of these features can make classification information diverse and avoid entangling with useless domain information. Since the diversity of class information is increased, the corresponding labels are also mixed for better matching, which is different from Mixstyle [14] and FACT [18]. As shown in Figure 1(b), when Mixup is performed on domain-invariant features that have no interference from classification information, the diversity of data is indeed enhanced with almost no unrecognizable synthetic data points being generated.

As shown in Figure 2, we adopt DANN [6] to learn domain-invariant features for its popularity and effectiveness. Nevertheless, FIX can also work with other methods for domain-invariant learning, which is shown in later experiments in Sec. 5.4[2]. The outputs of the bottleneck layer are viewed as domain-invariant features. The Mixup operation is performed on this layer. Correspondingly, the class labels are also mixed to increase the diversity of data while domain labels remain unchanged. Feature Mixup is performed within each batch. Concretely, for a batch of $\mathbf{z}$, we shuffle its indices and obtain $\hat{\mathbf{z}}$.

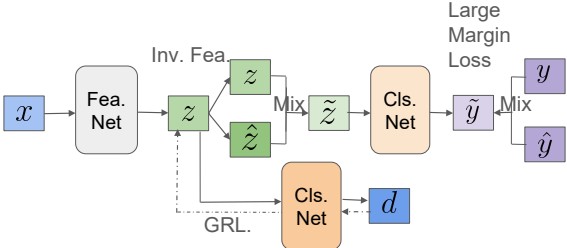

Figure 2: The network architecture of FIXED.

Then, $\mathbf{z}$ and $\hat{\mathbf{z}}$ are mixed to obtain $\tilde{\mathbf{z}}$, which is used as the input for the subsequent layers. At the same time, $\tilde{y}$ is generated accordingly.

## 3.2. Enhancing Discrimination

To further enhance discrimination, we introduce a *large margin* loss into Mixup just as in [20] to complete the design of FIXED. We follow [21] to derive the large margin loss as:

$$\ell_{lm}(h(\mathbf{x}_i), y_i) = \mathscr{G}_{k \neq y_i} \max\{0, \gamma + d_{h, \mathbf{x}_i, \{k, y_i\}} \text{sign}(h_k(\mathbf{x}_i) - h_{y_i}(\mathbf{x}_i)\}, \tag{3}$$

where $\ell_{lm}$ is large margin loss, $\mathscr{G}$ is an aggregation operator for the multi-class setting, and $\text{sign}(\cdot)$ adjusts the polarity of the distance. $h_k : \mathcal{X} \to \mathbb{R}$ generates a prediction score for classifying the input vector $\mathbf{x} \in \mathcal{X}$ to class $k$. $\gamma$ is the distance to the boundary that we expect. $d_{h, \mathbf{x}, \{k_1, k_2\}}$ is the distance of a point $\mathbf{x}$ to the decision boundary of class $i$ and $j$, which can be computed as:

$$d_{h, \mathbf{x}, \{k_1, k_2\}} = \min_{\delta} ||\delta||_p, s.t. \ h_{k_1}(\mathbf{x} + \delta) = h_{k_2}(\mathbf{x} + \delta), \tag{4}$$

where $|| \cdot ||_p$ is $l_p$ norm. As shown in [21], Eq. (3) can be computed as:

$$\mathscr{G}_{k \neq y_i} \max\{0, \gamma + \frac{h_k(\mathbf{x}_i) - h_{y_i}(\mathbf{x}_i)}{\|\nabla_{\mathbf{x}} h_k(\mathbf{x}_i) - \nabla_{\mathbf{x}} h_{y_i}(\mathbf{x}_i)\|_q}\}, \tag{5}$$

where $q = \frac{p}{p-1}$. As shown in Figure 1(d), a large margin can reduce noisy synthetic data points to enhance discrimination.

## 3.3. Summary

Combining the domain-invariant feature learning module and the large margin module, the objective function of FIXED can be formulated as:

$$\min \mathbb{E}_{(\mathbf{x}_1, y_1), (\mathbf{x}_2, y_2) \sim \mathbb{P}} \mathbb{E}_{\lambda \sim Beta(\alpha, \alpha)} [\ell_{lm}(G_y(\text{Mix}_\lambda(\mathbf{z}_1, \mathbf{z}_2)), \text{Mix}_\lambda(y_1, y_2)) + \ell_d(G_d(R_\eta(\mathbf{z}_1)), D)], \tag{6}$$

where $\mathbb{P}$ denotes the distribution of all data. $\text{Mix}(\cdot, \cdot)$ is a Mixup function, $\mathbf{z}_1 = G_f(\mathbf{x}_1)$, $\mathbf{z}_2 = G_f(\mathbf{x}_2)$ with $G_f, G_y, G_d$ the feature net, classification layer, and discriminator, respectively. We perform FIXED in batches. $\ell_d$ is the cross-entropy loss. $R_\eta$ is the gradient reversal layer [6] and $D$ is the domain label. Note that DANN is only one possible option for domain-invariant learning. We show that FIXED can work with CORAL [22] as an alternative implementation in Section 5.4.

# 4. Analytical Evaluation

In this section, we offer theoretical analysis to shed light on the reasons behind the remarkable performance of FIXED from two aspects: 1) distribution coverage and 2) inter-class distance.

---

[2]In the following, if there is no special note, FIX denotes FIX implemented with DANN.

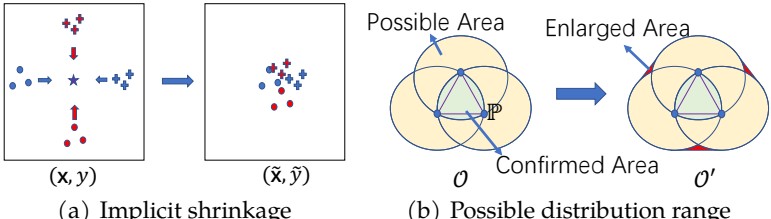

(a) Implicit shrinkage  (b) Possible distribution range

Figure 3: Toy examples of theoretical insights. (a) After the implicit shrinkage via Mixup, classes (denoted by different colors) mix together, bringing difficulty to classification. Different shapes denote domains. (b) FIXED enlarges the distribution cover range. Vertices represent distributions while colored areas represent possible $\mathcal{O}$ in Prop. C.2 and $\mathcal{O}'$ in Prop. 4.2.

## 4.1. Why Mixup is not Good Enough

**Proposition 4.1.** *(modified from Theorem 1 in [23]). Let $\theta \sim Beta_{[\frac{1}{2},1]}(\alpha, \alpha)$ and $j \sim Unif([n])$ be two random variables with $\alpha > 0$, $n > 0$, and let $\bar{\theta} = \mathbb{E}_\theta \theta$. For any training set $\mathcal{S}_n$, there exist two random perturbations $(\delta_i, \epsilon_i)$ with $\mathbb{E}_{\theta,j}\delta_i = \mathbb{E}_{\theta,j}\epsilon_i = 0, i \in [n]$. Denote $\varepsilon^{Mixup}$ the error of Mixup, we have*

$$\varepsilon^{Mixup}(h) = \frac{1}{n}\sum_{i=1}^{n}\mathbb{E}_{\theta,j}\ell(h(\tilde{\mathbf{x}}_i),\tilde{y}_i) = \frac{1}{n}\sum_{i=1}^{n}\mathbb{E}_{\theta,j}\ell(h(\bar{\mathbf{x}} + \bar{\theta}(\mathbf{x}_i - \bar{\mathbf{x}}) + \delta_i), \bar{y} + \bar{\theta}(y_i - \bar{y}) + \epsilon_i). \quad (7)$$

Note that $\bar{\theta} \in [1/2, 1]$. From Eq. (7), we can see that the transformation from $(\mathbf{x}_i, y_i)$ to $(\tilde{\mathbf{x}}_i, \tilde{y}_i)$ shrinks the inputs and the outputs towards their mean with perturbations. When there exist spurious relations induced by redundant domain information, it may bring confusion when performing Mixup, which is demonstrated in Figure 3(a). Moreover, introducing the large margin can make classes far from each other and thereby reduce confusion during Mixup.

## 4.2. FIXED has Larger Distribution Coverage

We derive our theory to prove that FIXED has a larger distribution coverage.

**Proposition 4.2.** *Let $\mathcal{X}$ be a space and $\mathcal{H}$ be a class of hypotheses corresponding to this space. Let $\mathbb{Q}$ and the collection $\{\mathbb{P}_i\}_{i=1}^M$ be distributions over $\mathcal{X}$ and let $\{\phi_i\}_{i=1}^M$ be a collection of non-negative coefficients with $\sum_i \phi_i = 1$. Let the object $\mathcal{O}'$ be a set of distributions such that for every $\mathbb{S} \in \mathcal{O}'$ the following holds*

$$d_{\mathcal{H}\Delta\mathcal{H}}(\sum_i \phi_i\mathbb{P}_i, \mathbb{S}) \leq \max_{i,j} d_{\mathcal{H}\Delta\mathcal{H}}(\mathbb{P}_i, \mathbb{P}_j). \quad (8)$$

*Then, for any $h \in \mathcal{H}$,*

$$\varepsilon_{\mathbb{Q}}(h) \leq \lambda' + \sum_i \phi_i \varepsilon_{\mathbb{P}_i}(h) + \frac{1}{2}\min_{\mathbb{S}\in\mathcal{O}'} d_{\mathcal{H}\Delta\mathcal{H}}(\mathbb{S}, \mathbb{Q}) + \frac{1}{2}\max_{i,j} d_{\mathcal{H}\Delta\mathcal{H}}(\mathbb{P}_i, \mathbb{P}_j) \quad (9)$$

*where $\lambda'$ is the error of an ideal joint hypothesis.*

**Proposition 4.3.** *Under the same conditions in 4.2,*

$$\mathcal{O} = \{S| \sum_i \phi_i d_{\mathcal{H}\Delta\mathcal{H}}(\mathbb{P}_i, \mathbb{S}) \leq \max_{i,j} d_{\mathcal{H}\Delta\mathcal{H}}(\mathbb{P}_i, \mathbb{P}_j)\}, \quad (10)$$

$$\mathcal{O}' = \{S| d_{\mathcal{H}\Delta\mathcal{H}}(\sum_i \phi_i\mathbb{P}_i, \mathbb{S}) \leq \max_{i,j} d_{\mathcal{H}\Delta\mathcal{H}}(\mathbb{P}_i, \mathbb{P}_j)\}, \quad (11)$$

*we have*

$$\mathcal{O} \subset \mathcal{O}'. \quad (12)$$

From Prop. 4.2 and Prop. 4.3[3], $\mathcal{O}'$ has a larger possible cover range than $\mathcal{O}$, which brings more *diversity*. As shown in the right part of Figure 3(b), the red area is the possible increased area. Possible

---

[3]Proofs can be found in section C.2.

areas contain distributions that may be in $\mathcal{O}(\mathcal{O}')$ while confirmed areas contain distributions that must be in $\mathcal{O}(\mathcal{O}')$. It reveals that the area with a constant distance to purple lines and blue vertices is larger than the area with the same distance to blue vertexes, where purple lines can be viewed as Mixup of vertexes expressed as $\sum_i \phi_i \mathbb{P}_i$. Moreover, yellow areas may have a higher possibility to satisfy Eq. (8) since the points in it have shorter distances from purple lines and blue vertices.

### 4.3. Insights from Inter-class and Intra-class Distances

Recently, it has come to researchers' attention that just learning domain invariant features may not be enough for good generalization and discrimination, especially in the field of domain adaptation [13, 24]. A common approach to enhance generalization and discrimination is to enlarge the inter-class distance and decrease the intra-class distance [25], which has already been utilized for domain adaptation [26] and domain generalization [27]. From Eq. 7, feature Mixup can be perceived as a tool to decrease the intra-class distance, while the large margin loss can enlarge the inter-class distance. This indicates that FIXED enhances generalization and discrimination.

## 5. Experimental Evaluation

While most literature on DG evaluates the algorithms on image classification datasets, we perform evaluations on **both** image classification and sensor-based human activity recognition (i.e., time series) data. This can help study the generality of our method across multiple modalities.

### 5.1. Evaluation on Image Classification Datasets

#### 5.1.1. Datasets

We adopt three popular DG benchmark datasets. (1) **Digits-DG** [28], which contains four digit datasets including MNIST [29], MNIST-M [30], SVHN [31], SYN [29]. The four datasets differ in font style, background, and image quality. Following [28], we select 600 images per class from each dataset. (2) **PACS** [32], which is an object classification benchmark with four domains (i.e., photos, art-paintings, cartoons, sketches). There exist large discrepancies in image styles among different domains. Each domain contains seven classes and there are 9,991 images in total. (3) **Office-Home** [33], which is an object classification benchmark that contains four domains (i.e., Art, Clipart, Product, Real-World). The domain shift comes from image styles and viewpoints. Each domain contains 65 classes and there are 15,500 images in total.

#### 5.1.2. Baselines and Implementation Details

For the experiments using ResNet-18, i.e., Office-Home and PACS datasets, and Digits-DG dataset that uses DTN as the backbone following [34], we re-implement several recent strong comparison methods by extending the DomainBed [35] codebase for fair study. For the algorithms that are not implemented by ourselves, we copy their results from their papers when the settings are the same. Our reproductions are marked with *. We select the best model via results on validation datasets. Specifically, we split each source domain with a ratio of $8:2$ for training and validation following DomainBed and report average results of three trials[4].

#### 5.1.3. Results and Discussion

Table 1 shows the results on PACS, Office-Home, and Digits-DG datasets respectively where PACS and Office-Home used ResNet-18. We observe that FIXED consistently outperforms all comparison methods. For PACS, our method can have an over $1\%$ improvement compared to the second-best one. In an absolutely fair environment, our method can even achieve an over $3\%$ improvement compared to the methods with stars. $3\%$ is a remarkable improvement since some methods, e.g. DANN, only have slight improvements compared to ERM. Our method can also have over $1\%$ and $0.3\%$ improvements compared to the second-best ones for Office-Home and Digits-DG respectively.

---

[4]The ratio $8:2$ is suggested by DomainBed for fairness. Several methods adopted $9:1$ which involves more training data that are easier to perform better. In these cases, our method still outperforms them.

Table 1: The results on PACS, Office-Home, and Digits-DG. The bold items are the best results.

| Method | PACS | | | | | Method | Office-Home | | | | |
|---|---|---|---|---|---|---|---|---|---|---|---|
| | A | C | P | S | AVG | | A | C | P | R | AVG |
| ERM* | 77 | 74.53 | 95.51 | 77.86 | 81.22 | ERM* | 58.63 | 47.95 | 72.22 | 73.03 | 62.96 |
| DANN* [6] | 78.71 | 75.3 | 94.01 | 77.83 | 81.46 | DANN* [6] | 57.73 | 44.42 | 71.95 | 72.5 | 61.65 |
| CORAL* [22] | 77.78 | 77.05 | 92.63 | 80.55 | 82 | CORAL* [22] | 58.76 | 48.75 | 72.34 | 73.63 | 63.37 |
| Mixup* [1] | 79.1 | 73.46 | 94.49 | 76.71 | 80.94 | MMD-AAE [11] | 56.5 | 47.3 | 72.1 | **74.8** | 62.7 |
| MetaReg [12] | 83.7 | 77.2 | 95.5 | 70.3 | 81.7 | Mixup* [1] | 55.79 | 47.88 | 71.95 | 72.83 | 62.11 |
| Jigen [36] | 79.42 | 75.25 | 96.03 | 71.35 | 80.51 | Jigen [36] | 53.04 | 47.51 | 71.47 | 72.79 | 61.2 |
| Epi-FCR [37] | 82.1 | 77 | 93.9 | 73 | 81.5 | MIX-ALL* [17] | 56.08 | 46.9 | 72.07 | 73.93 | 62.24 |
| GroupDRO* [38] | 76.03 | 76.07 | 91.2 | 79.05 | 80.59 | GroupDRO* [38] | 57.6 | 48.77 | 71.53 | 73.17 | 62.77 |
| RSC* [39] | 79.74 | 76.11 | 95.57 | 76.64 | 82.01 | RSC* [39] | 58.96 | 49.16 | 72.54 | 74.16 | 63.7 |
| MIX-ALL* [17] | 80.66 | 73.85 | 93.83 | 76.05 | 81.1 | ANDMask* [40] | 56.74 | 45.86 | 70.67 | 73.19 | 61.61 |
| L2A-OT [41] | 83.3 | 78.2 | 96.2 | 73.6 | 82.8 | SagNet [42] | 60.2 | 45.38 | 70.42 | 73.38 | 62.34 |
| MMLD [43] | 81.28 | 77.16 | 96.09 | 72.29 | 81.83 | Ours | **61.06** | **50.08** | **73.39** | 74.45 | **64.75** |

| Method | PACS | | | | | Method | Digits-DG | | | | |
|---|---|---|---|---|---|---|---|---|---|---|---|
| DDAIG [28] | 84.2 | 78.1 | 95.3 | 74.7 | 83.1 | | MNIST | MNIST-M | SVHN | SYN | AVG |
| SNR [44] | 80.3 | 78.2 | 94.5 | 74.1 | 81.8 | ERM* | 97.55 | 55.52 | 59.98 | 89.25 | 75.58 |
| EISNet [45] | 81.89 | 76.44 | 95.93 | 74.33 | 82.15 | DANN* [6] | 97.77 | 55.62 | 61.85 | 89.37 | 76.15 |
| CSD [46] | 78.9 | 75.8 | 94.1 | 76.7 | 81.4 | CORAL* [22] | 97.62 | 57.68 | 57.82 | 90.12 | 75.81 |
| InfoDrop [47] | 80.27 | 76.54 | 96.11 | 76.38 | 82.33 | MMD-AAE [11] | 96.5 | 58.4 | **65** | 78.4 | 74.6 |
| ANDMask* [40] | 76.22 | 73.81 | 91.56 | 78.06 | 79.91 | Mixup* [1] | 97.5 | 57.95 | 54.75 | 89.8 | 75 |
| CuMix [48] | 82.3 | 76.5 | 95.1 | 72.6 | 81.6 | Jigen [36] | 96.5 | 61.4 | 63.7 | 74 | 73.9 |
| StableNet [49] | 81.74 | 79.91 | **96.53** | 80.5 | 84.69 | GroupDRO* [38] | 97.48 | 53.47 | 55.63 | **92.15** | 74.68 |
| MixStyle [14] | 84.1 | 78.8 | 96.1 | 75.9 | 83.7 | RSC* [39] | **97.78** | 56.27 | 62.38 | 89.25 | 76.42 |
| SagNet [42] | 83.58 | 77.66 | 95.47 | 76.3 | 83.25 | MIX-ALL* [17] | 96.23 | 59.28 | 54.73 | 83.57 | 73.45 |
| MatchDG [50] | 79.77 | **80.03** | 95.93 | 77.11 | 83.21 | MixStyle [14] | 96.5 | **63.5** | 64.7 | 81.2 | 76.5 |
| L2D [51] | 81.44 | 79.56 | 95.51 | 80.58 | 84.27 | ANDMask* [40] | 96.85 | 56 | 59.47 | 88.17 | 75.12 |
| SFA [52] | 81.2 | 77.8 | 93.9 | 73.7 | 81.7 | Ours | 97.67 | 56.72 | 64.17 | 88.73 | **76.82** |
| Ours | **84.23** | 78.8 | 96.37 | **83.54** | **85.74** | | | | | | |

We observe more insightful conclusions. (1) Vanilla Mixup even performs worse than ERM on some benchmarks, which illustrates that mixed domain information interfaces performance seriously. (2) There are small performance gaps among different methods on some benchmarks, e.g. Office-Home; and even ERM can achieve acceptable results. It is caused because there are few differences among different domains (Office-Home) or some other reasons. (3) Different domain splits are important for DG. For example, in Digits-DG, MixStyle shows a significant improvement on the second task. Even Jigen performs better than all other methods. Hence, it is important to perform several random trials to record the average performance.

## 5.2. Evaluation on Human Activity Recognition

### 5.2.1. Datasets and Settings

We evaluate our method on several DG benchmarks with three different settings on human activity recognition. Four datasets are used: UCI daily and sports dataset (DSADS) [53], USC-SIPI human activity dataset (USC-HAD) [54], UCI human activity recognition using smartphones data set (UCI-HAR) [55], and PAMAP2 physical activity monitoring dataset (PAMAP2) [56]. We mainly use the sliding window technique to preprocess data. We constructed three settings for extensive evaluations of our method: (1) **Cross-Person**: This setting utilizes USC-HAD dataset and 14 persons are divided into four groups. Each domain contains 12 classes. Each sample has two sensors with six dimensions. (2) **Cross-Position**: This setting utilizes DSADS dataset and data from each position corresponds to a different domain. There are 19 classes in total. Each sample contains three sensors with nine dimensions. (3) **Cross-Dataset**: This setting utilizes all four datasets, and each dataset corresponds to a different domain. Data of six common classes, including walking, walking upstairs, walking downstairs, sitting, standing, and laying, are selected. We choose two sensors from each dataset that belong to the same position. Data is down-sampled to ensure the dimensions of data in different datasets same. It is easy to see that cross-dataset setting is more challenging than the other two since it contains more diversity.

### 5.2.2. Baselines and Implementation Details

For all three settings, we reproduce eight state-of-the-art baselines by ourselves. In addition, for cross-person on USC-HAD, we add the results of GILE [57] obtained from results via their code. For each benchmark, we randomly split each source domain into 80% for training and 20% for validation.

Table 2: Results on HAR benchmarks. The **bold** and underline mean the best and second-best.

| | Source | Target | ERM* | DANN* [6] | CORAL* [22] | ANDMask* [40] | GroupDRO* [38] | RSC* [39] | Mixup* [1] | MIX-ALL* [17] | GILE [57] | FIXED |
|---|---|---|---|---|---|---|---|---|---|---|---|---|
| X-Person | 1,2,3 | 0 | 80.98 | 81.22 | 78.82 | 79.88 | 80.12 | 81.88 | 79.98 | 78.44 | 78.00 | **84.73** |
| | 0,2,3 | 1 | 57.75 | 57.88 | 58.93 | 55.32 | 55.51 | 57.94 | 64.14 | 59.32 | 62.00 | **67.90** |
| | 0,1,3 | 2 | 74.03 | 76.69 | 75.02 | 74.47 | 74.69 | 73.39 | 74.32 | 72.96 | 77.00 | **79.21** |
| | 0,1,2 | 3 | 65.86 | 70.72 | 53.72 | 65.04 | 59.97 | 65.13 | 61.28 | 63.46 | 63.00 | **74.47** |
| | AVG | - | 69.66 | 71.63 | 66.62 | 68.68 | 67.57 | 69.59 | 69.93 | 68.54 | 70.00 | **76.58** |

| | Source | Target | ERM* | DANN* [6] | CORAL* [22] | ANDMask* [40] | GroupDRO* [38] | RSC* [39] | Mixup* [1] | MIX-ALL* [17] | - | FIXED |
|---|---|---|---|---|---|---|---|---|---|---|---|---|
| X-Position | 1,2,3,4 | 0 | 41.52 | 45.45 | 33.22 | 47.51 | 27.12 | 46.56 | 48.77 | 40.1 | - | **49.57** |
| | 0,2,3,4 | 1 | 26.73 | 25.36 | 25.18 | 31.06 | 26.66 | 27.37 | 34.19 | 31.16 | - | **35.64** |
| | 0,1,3,4 | 2 | 35.81 | 38.06 | 25.81 | 39.17 | 24.34 | 35.93 | 37.49 | 41.16 | - | **40.44** |
| | 0,1,2,4 | 3 | 21.45 | 28.89 | 22.32 | 30.22 | 18.39 | 27.04 | 29.50 | 30.56 | - | **33.00** |
| | 0,1,2,3 | 4 | 27.28 | 25.05 | 20.64 | 29.90 | 24.82 | 29.82 | 29.95 | 29.31 | - | **33.22** |
| | AVG | - | 30.56 | 32.56 | 25.43 | 35.57 | 24.27 | 33.34 | 35.98 | 34.46 | - | **38.37** |

| | Source | Target | ERM* | DANN* [6] | CORAL* [22] | ANDMask* [40] | GroupDRO* [38] | RSC* [39] | Mixup [1] | MIX-ALL* [17] | - | FIXED |
|---|---|---|---|---|---|---|---|---|---|---|---|---|
| X-Dataset | 1,2,3 | 0 | 26.35 | 29.73 | 39.46 | 41.66 | **51.41** | 33.10 | 37.35 | 31.01 | - | 46.73 |
| | 0,2,3 | 1 | 29.58 | 45.33 | 41.82 | 33.83 | 36.74 | 39.70 | 47.39 | 38.42 | - | **53.28** |
| | 0,1,3 | 2 | 44.44 | 46.06 | 39.10 | 43.22 | 33.20 | 45.28 | 40.24 | 37.52 | - | **46.15** |
| | 0,1,2 | 3 | 32.93 | 43.84 | 36.61 | 40.17 | 33.80 | 45.94 | 23.12 | 22.8 | - | **53.67** |
| | AVG | - | 33.32 | 41.24 | 39.25 | 39.72 | 38.79 | 41.01 | 37.03 | 32.44 | - | **49.96** |
| - | AVG | - | 44.51 | 48.48 | 43.77 | 47.99 | 43.54 | 47.98 | 47.65 | 45.15 | - | **54.97** |

### 5.2.3. Results and Discussion

The results on time series are presented in Table 2. Overall, our method has an improvement of **6.5**% average accuracy than the second-best method on average of these three settings. This demonstrates generalization capability across different tasks of our method. These settings represent DG scenarios of different difficulties

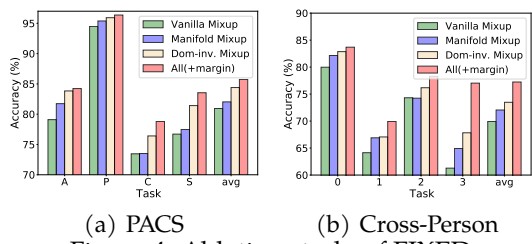

(a) PACS      (b) Cross-Person
Figure 4: Ablation study of FIXED.

(e.g., cross-dataset is much more difficult than cross-person), thus they can thoroughly reflect the performance of all methods in different situations. As shown in Table 2, some methods deteriorate seriously for these three HAR benchmarks while our method achieves the best average accuracy on average and performs best almost on every task. The results are consistent with image datasets. To sum up, our method is effective in both image and time series datasets, indicating that it is a general approach for domain generalization.

## 5.3. Qualitative Analysis

### 5.3.1. Ablation Study

We report our ablation study in Figure 4. Compared with vanilla Mixup, Manifold Mixup shows slight improvements. It is caused by the reason that the model trained with categories as goals is biased towards containing more classification information in the deeper layers[5]. It proves our motivation of domain-

Table 3: Ablation study of FIXED (PACS) to show the improvement on large margin loss.

| ALG | A | C | P | S | AVG |
|---|---|---|---|---|---|
| ERM | 77.00 | 74.53 | 95.51 | 77.86 | 81.22 |
| ERM+Margin | 81.98 | 74.87 | 96.17 | 76.81 | 82.46 |
| DANN | 78.71 | 75.30 | 94.01 | 77.83 | 81.46 |
| DANN+Margin | 82.52 | 76.62 | 95.57 | 78.49 | 83.30 |
| Mixup | 79.10 | 73.46 | 94.49 | 76.71 | 80.94 |
| Mixup+Margin | 82.81 | 74.23 | 94.91 | 81.32 | 83.32 |
| Ours | **84.23** | **78.8** | **96.37** | **83.54** | **85.74** |

invariant Mixup from another view. Compared with Manifold Mixup, it is obvious that directly using Mixup on domain-invariant features has a remarkable improvement, which demonstrates that increasing the diversity of useful information for model training is able to bring benefits. In addition, we present results on large margin with existing methods in Table 3 After introducing large margin, there are extra improvements compared with domain-invariant Manifold Mixup. These experiments demonstrate that the two components are both effective.

### 5.3.2. Visualization Study

We present visualization results to show the rationale of our method. As shown in Figure 5(a), the same class in different domains has different distributions with ERM (data points with the same color and different shapes locate in different places), which is just like Figure 1(a). Moreover, some classes are close to each other, which is like Figure 1(c). If we do nothing for these situations, synthetic noisy points will be generated as mentioned above. Even if vanilla Mixup or Manifold Mixup

---

[5]We try our best to perform Manifold Mixup in the deeper layers, which leads to remarkable improvements compared to Vanilla Mixup, but it is still worse than ours.

is used, the above issues cannot be solved (see Figure 5(b) and Figure 5(c) respectively). With domain invariant features, FIXED can reduce the influence generated by redundant domain information (Figure 5(d)) while it can enhance discrimination. Overall, with both considerations, FIXED achieves the best visualization effects in Figure 5(f) where two problems have been relieved to a certain degree and thus leads to the best performance in Figure 4.

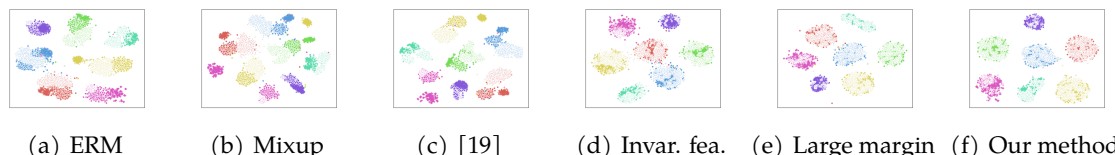

(a) ERM     (b) Mixup     (c) [19]     (d) Invar. fea.     (e) Large margin     (f) Our method

Figure 5: Visualization of the t-SNE embeddings of learned feature spaces for PACS with different methods. Different colors correspond to different classes and different shapes correspond to different domains. *Best viewed in color and zoom in.*

## 5.4. More Analysis

### 5.4.1. Extensibility

To demonstrate that FIX is extensible, we replace the adversarial learning module in FIX with CORAL loss [22] to learn domain-invariant features. For better comparison, we do not use large margin and we denote it as FIX-CORAL. We compare it with CORAL and Vanilla Mixup on two benchmarks: Office-Home and Cross-Person. From Figure 6(a) and Figure 6(b), we can see FIX-CORAL achieves the best results on both benchmarks compared with CORAL and vanilla Mixup, which demonstrates FIXED is a general approach for domain generation. In addition, in Figure 6(a), it can be observed that FIX-CORAL without large margin loss almost achieves the same performance as FIXED with large margin loss. It may be caused by that CORAL performs better than adversarial training on Office-Home (rf. Table 1), thereby obtaining improved domain-invariant features.

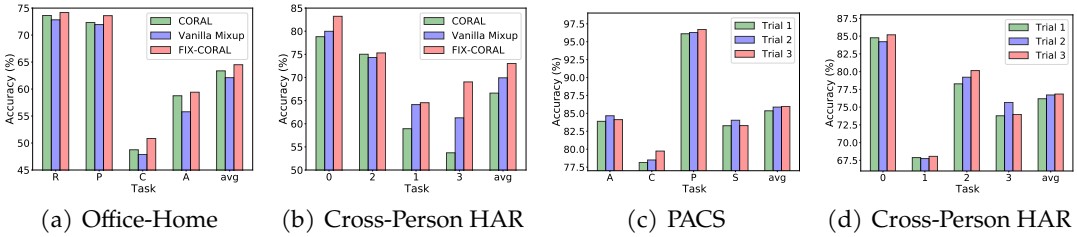

(a) Office-Home     (b) Cross-Person HAR     (c) PACS     (d) Cross-Person HAR

Figure 6: Figure 6(a)-6(b) are results of FIXED with CORAL extension while Figure 6(c)-6(d) are results on cross-person HAR and PACS with three trials to show the robustness of our algorithm.

### 5.4.2. Robustness

Our method involves Mixup strategy and random domain splits which may introduce instabilities. In this section, we evaluate its robustness. Figure 6(c) and Figure 6(d) demonstrate that our method is robust against random seeds and different domain splits in several trials. This implies that our method can be easily applied to real applications.

## 6. Conclusions and Future Work

In this paper, we proposed FIXED, a general approach for domain generalization. FIXED performs Mixup on domain-invariant features to increase diversity by discerning domain and class information. To mitigate the noisy synthetic data problem in Mixup, we introduced the large margin loss. FIXED can be embedded in many existing DG methods, and we presented implementations based on DANN and CORAL. We provided theoretical insights to our algorithm. Extensive experiments on seven datasets across two modalities demonstrated that FIXED yielded SOTA results on all datasets. In the future, we plan to incorporate our approach into representation learning and meta-learning DG methods to improve performance and deploy it on more applications.

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

## A. Broader Impact Concerns

FIXED introduces innovative data augmentation techniques to enhance machine learning model robustness and generalization. However, we must consider its real-world implications, including data privacy issues, potential biases from training data, and fairness concerns in model outcomes. Evaluating its benefits in diverse real-world scenarios and addressing its susceptibility to adversarial attacks are essential. By holistically assessing these concerns, we aim to ensure the responsible and ethical application of FIXED, maximizing its positive impact while minimizing adverse consequences.

## B. Related Work

### B.1. Domain Generalization

DG aims to learn a model from single or multiple source domains to generalize well to unseen target domains. According to [10], existing DG methods can be divided into three groups: 1) representation learning [44, 58, 59], 2) learning strategy [60, 61], and 3) data manipulation [28, 62–64].

Representation learning is one of the most common approaches for domain adaptation and domain generalization. Since DG can be viewed as an extension of DA, many traditional DA methods can be applied to DG [6, 11, 22]. Domain-adversarial neural network (DANN) [6] learns domain-invariant features via adversarial training of the generator and the discriminator. The discriminator aims to distinguish the domains, while the generator aims to fool the discriminator to learn domain invariant feature representations. Deep Coral [22] learns a nonlinear transformation that aligns correlations of layer activations in deep neural networks. MMD-AAE [11] imposes the Maximum Mean Discrepancy (MMD) measure to align the distributions among different domains, and matches the aligned distribution to an arbitrary prior distribution via adversarial feature learning. These methods all attempt to learn feature representations that are supposed to be universal to the seen source domains, and are expected to generalize well on the target domain. There are also other methods for domain-invariant learning on DG [50, 65]. Recent research works point out that just learning domain-invariant feature representation may be not enough for DA [66] and DG [13]. Therefore, [13] proposed a novel theoretically sound framework - mDSDI - to further capture the usefulness

of domain-specific information. The proposed FIXED method is another approach to exploit additional information beyond the domain-invariant feature representation.

Learning strategy is another group for DG approaches. In this group, methods focus on exploiting the general learning strategy to enhance model generalization capability. MLDG [9] proposed a model agnostic training procedure for DG that simulates train/test domain shift during training by synthesizing virtual testing domains within each mini-batch. The meta-optimization objective requires that steps to improve training domain performance must also improve test domain performance. Fish [61] utilizes an inter-domain gradient matching objective that targets domain generalization by maximizing the inner product between gradients from different domains. By forcing the gradient direction to be invariant for different domains, it can be generalized to unseen targets. Similar to Fish, [67] conjectured that conflicting gradients within each mini-batch contain information specific to the individual domains which are irrelevant to others when training with multiple domains. It characterizes the conflicting gradients and devises novel gradient agreement strategies based on gradient surgery to alleviate such disagreements to improve generalization.

The data manipulation group of approaches is the most closely related to our work. Thus, we elaborate on it in the next subsection.

## B.2. Data Augmentation and Mixup for DG

Data augmentation is used in DG through either domain randomization [68], self-supervised learning (e.g., JiGen [36]), adversarial augmentation (e.g., CrossGrad [69]), or Mixup [1]. In [68], an approach for domain randomization and pyramid consistency is proposed to learn a model with high generalizability. In particular, it randomizes the synthetic images with the styles of real images in terms of visual appearance using auxiliary datasets. JeGen [36] learns the semantic labels in a supervised fashion, and broadens its understanding of the data by learning from self-supervised signals to solve a jigsaw puzzle on the same images. This helps the network to learn the concepts of spatial correlation while acting as a regularizer for the classification task. CrossGrad [69] trains a label and a domain classifier in parallel on examples perturbed by loss gradients of each other's objectives. Recently, SFA [52], a simple feature augmentation method, attempts to perturb the feature embedding with Gaussian noise during training.

Mixup [1] is a simple but effective technique to increase data diversity. It extends the training distribution by incorporating the intuition that linear interpolations of feature vectors shall lead to linear interpolations of the associated targets. There are several variants of Mixup. Manifold Mixup [19] is designed to encourage neural networks to predict less confidently on interpolations of hidden representations, which leverages semantic interpolations as additional training signals. Note that manifold Mixup is a common approach while both the vanilla Mixup and FIX can be regarded as specific implementations of manifold Mixup for different purposes. CutMix [70] replaces the removed regions with a patch from another image and mixes the ground truth labels proportionally to the number of pixels of combined images. Puzzle Mix [71] explicitly utilizes the saliency information and the underlying statistics of the natural examples to prevent misleading supervisory signals.

Since Mixup is a natural way for data augmentation, many variants of Mixup have been proposed for DA and DG. Recent works [72, 73] use the vanilla Mixup method for domain adaptation without modification. DM-ADA [72], a Mixup-based method for DA, utilizes domain mixup on the pixel level and the feature level to improve model robustness. It guarantees domain-invariance in a more continuous latent space, and guides the domain discriminator to judge sample difference between the source and target domains. Mixstyle [14] increases the image style information to enhance the diversity of domains without changing classification labels. FACT [18] mixes the amplitude spectrum of images after Fourier transform to force the model to capture phase information. Wang et al. [17] mixes up samples in multiple domains with two different sampling strategies.

Existing methods generally ignore the domain-invariant features and discrimination of Mixup. In addition, they are designed for specific tasks and need to be modified for each application domain. FIXED can address these limitations.

# C. Analytical Evaluation

## C.1. Background

For a distribution $\mathbb{P}$ with an ideal binary labeling function $h^*$ and a hypothesis $h$, we define the error $\varepsilon_{\mathbb{P}}(h)$ in accordance with [74] as:

$$\varepsilon_{\mathbb{P}}(h) = \mathbb{E}_{\mathbf{x} \sim \mathbb{P}} |h(\mathbf{x}) - h^*(\mathbf{x})|. \tag{13}$$

The error of Mixup is given as [23]:

$$\varepsilon^{\text{Mixup}}(h) = \frac{1}{n^2} \sum_{i=1}^{n} \sum_{j=1}^{n} \mathbb{E}_\lambda \ell(h(\lambda \mathbf{x}_i + (1-\lambda)\mathbf{x}_j), \lambda y_i + (1-\lambda)y_j), \tag{14}$$

where $\lambda \sim Beta(\alpha, \alpha)$ and $n$ is the number of data samples.

We also give the definition of $\mathcal{H}$-divergence in accordance with [74]. Given two distributions $\mathbb{P}, \mathbb{Q}$ over a space $\mathcal{X}$ and a hypothesis class $\mathcal{H}$,

$$d_{\mathcal{H}}(\mathbb{P}, \mathbb{Q}) = 2 \sup_{h \in \mathcal{H}} |Pr_{\mathbb{P}}(I_h) - Pr_{\mathbb{Q}}(I_h)|, \tag{15}$$

where $I_h = \{\mathbf{x} \in \mathcal{X} | h(\mathbf{x}) = 1\}$. We often consider the $\mathcal{H}\Delta\mathcal{H}$-divergence in [74] where the symmetric difference hypothesis class $\mathcal{H}\Delta\mathcal{H}$ is the set of functions characterized by disagreements between hypotheses.

For any $n \in \mathbb{N}, [n] = \{1, \cdots, n\}$ is the set of nonzero integers up to $n$. For any $\alpha, \beta > 0, [a, b] \subset [0, 1]$, $Beta_{[a,b]}(\alpha, \beta)$ denotes the truncated Beta distribution on $[a, b]$. $j \sim Unif([n])$ represents uniform random sampling.

**Theorem C.1.** (*Theorem 2.1 in* [75], *modified from Theorem 2 in* [74]). *Let $\mathcal{X}$ be a space and $\mathcal{H}$ be a class of hypotheses corresponding to this space. Suppose $\mathbb{P}$ and $\mathbb{Q}$ are distributions over $\mathcal{X}$. Then, for any $h \in \mathcal{H}$, the following holds*

$$\varepsilon_{\mathbb{Q}}(h) \leq \lambda'' + \varepsilon_{\mathbb{P}}(h) + \frac{1}{2} d_{\mathcal{H}\Delta\mathcal{H}}(\mathbb{Q}, \mathbb{P}) \tag{16}$$

*with $\lambda''$ the error of an ideal joint hypothesis for $\mathbb{Q}$ and $\mathbb{P}$.*

Theorem C.1 provides an upper bound on the target-error. $\lambda''$ is a property of the dataset and hypothesis class and is often ignored. Theorem C.1 demonstrates the necessity to learn domain invariant features.

**Proposition C.2.** (*Proposition 3.1 in* [75], *modified from Proposition 2 in* [76]). *Let $\mathcal{X}$ be a space and $\mathcal{H}$ be a class of hypotheses corresponding to this space. Let $\mathbb{Q}$ and the collection $\{\mathbb{P}_i\}_{i=1}^{M}$ be distributions over $\mathcal{X}$ and let $\{\phi_i\}_{i=1}^{M}$ be a collection of non-negative coefficient with $\sum_i \phi_i = 1$. Let the object $\mathcal{O}$ be a set of distributions such that for every $\mathbb{S} \in \mathcal{O}$ the following holds*

$$\sum_i \phi_i d_{\mathcal{H}\Delta\mathcal{H}}(\mathbb{P}_i, \mathbb{S}) \leq \max_{i,j} d_{\mathcal{H}\Delta\mathcal{H}}(\mathbb{P}_i, \mathbb{P}_j). \tag{17}$$

*Then, for any $h \in \mathcal{H}$,*

$$\varepsilon_{\mathbb{Q}}(h) \leq \lambda_\phi + \sum_i \phi_i \varepsilon_{\mathbb{P}_i}(h) + \frac{1}{2} \min_{\mathbb{S} \in \mathcal{O}} d_{\mathcal{H}\Delta\mathcal{H}}(\mathbb{S}, \mathbb{Q}) + \frac{1}{2} \max_{i,j} d_{\mathcal{H}\Delta\mathcal{H}}(\mathbb{P}_i, \mathbb{P}_j) \tag{18}$$

*where $\lambda_\phi = \sum_i \phi_i \lambda_i$ and each $\lambda_i$ is the error of an ideal joint hypothesis for $\mathbb{Q}$ and $\mathbb{P}_i$.*

Proposition C.2 gives an upper bound for domain generalization. In the right-hand side of Eq. 18, the first term can be ignored and the second term is a convex combination of the source errors. The third term demonstrates the importance of diverse source distributions so that the unseen target $\mathbb{Q}$ might be near $\mathcal{O}$ while the final term is a maximum over the source-source divergences. As shown in Figure 3(b), the area with yellow and green is the possible $\mathcal{O}$.

## C.2. FIXED has Larger Distribution Coverage

We provide proofs of propositions here.

**Proposition C.3.** *Let $\mathcal{X}$ be a space and $\mathcal{H}$ be a class of hypotheses corresponding to this space. Let $\mathbb{Q}$ and the collection $\{\mathbb{P}_i\}_{i=1}^M$ be distributions over $\mathcal{X}$ and let $\{\phi_i\}_{i=1}^M$ be a collection of non-negative coefficient with $\sum_i \phi_i = 1$. Let the object $\mathcal{O}'$ be a set of distributions such that for every $\mathbb{S} \in \mathcal{O}'$ the following holds*

$$d_{\mathcal{H}\Delta\mathcal{H}}(\sum_i \phi_i \mathbb{P}_i, \mathbb{S}) \leq \max_{i,j} d_{\mathcal{H}\Delta\mathcal{H}}(\mathbb{P}_i, \mathbb{P}_j). \tag{19}$$

*Then, for any $h \in \mathcal{H}$,*

$$\varepsilon_{\mathbb{Q}}(h) \leq \lambda' + \sum_i \phi_i \varepsilon_{\mathbb{P}_i}(h) + \frac{1}{2} \min_{\mathbb{S} \in \mathcal{O}'} d_{\mathcal{H}\Delta\mathcal{H}}(\mathbb{S}, \mathbb{Q}) + \frac{1}{2} \max_{i,j} d_{\mathcal{H}\Delta\mathcal{H}}(\mathbb{P}_i, \mathbb{P}_j) \tag{20}$$

*where $\lambda'$ is the error of an ideal joint hypothesis.*

*Proof.* On one hand, with Theorem C.1, we have

$$\varepsilon_{\mathbb{Q}}(h) \leq \lambda_1 + \varepsilon_{\mathbb{S}}(h) + \frac{1}{2} d_{\mathcal{H}\Delta\mathcal{H}}(\mathbb{S}, \mathbb{Q}), \forall h \in \mathcal{H}, \forall \mathbb{S} \in \mathcal{O}'. \tag{21}$$

On the other hand, with Theorem C.1, we have

$$\varepsilon_{\mathbb{S}}(h) \leq \lambda_2 + \varepsilon_{\sum_i \phi_i \mathbb{P}_i}(h) + \frac{1}{2} d_{\mathcal{H}\Delta\mathcal{H}}(\sum_i \phi_i \mathbb{P}_i, \mathbb{S}), \forall h \in \mathcal{H}. \tag{22}$$

Since $\varepsilon_{\sum_i \phi_i \mathbb{P}_i}(h) = \sum_i \phi_i \varepsilon_{\mathbb{P}_i}(h)$, and $d_{\mathcal{H}\Delta\mathcal{H}}(\sum_i \phi_i \mathbb{P}_i, \mathbb{S}) \leq \max_{i,j} d_{\mathcal{H}\Delta\mathcal{H}}(\mathbb{P}_i, \mathbb{P}_j)$, we have

$$\varepsilon_{\mathbb{Q}}(h) \leq \lambda' + \sum_i \phi_i \varepsilon_{\mathbb{P}_i}(h) + \frac{1}{2} d_{\mathcal{H}\Delta\mathcal{H}}(\mathbb{S}, \mathbb{Q}) + \frac{1}{2} \max_{i,j} d_{\mathcal{H}\Delta\mathcal{H}}(\sum_i \phi_i \mathbb{P}_i, \mathbb{S}), \forall h \in \mathcal{H}, \forall \mathbb{S} \in \mathcal{O}'. \tag{23}$$

Eq. (23) for all $\mathbb{S} \in \mathcal{O}'$ holds. Proof ends. $\square$

**Proposition C.4.** *Under the same conditions in C.3,*

$$\mathcal{O} = \{S | \sum_i \phi_i d_{\mathcal{H}\Delta\mathcal{H}}(\mathbb{P}_i, \mathbb{S}) \leq \max_{i,j} d_{\mathcal{H}\Delta\mathcal{H}}(\mathbb{P}_i, \mathbb{P}_j)\}, \tag{24}$$

$$\mathcal{O}' = \{S | d_{\mathcal{H}\Delta\mathcal{H}}(\sum_i \phi_i \mathbb{P}_i, \mathbb{S}) \leq \max_{i,j} d_{\mathcal{H}\Delta\mathcal{H}}(\mathbb{P}_i, \mathbb{P}_j)\}, \tag{25}$$

*we have*

$$\mathcal{O} \subset \mathcal{O}'. \tag{26}$$

*Proof.* On one hand, for any $\mathbb{S} \in \mathcal{O}$, we have

$$\sum_i \phi_i d_{\mathcal{H}\Delta\mathcal{H}}(\mathbb{P}_i, \mathbb{S}) \leq \max_{i,j} d_{\mathcal{H}\Delta\mathcal{H}}(\mathbb{P}_i, \mathbb{P}_j). \tag{27}$$

On the other hand, with the triangle inequality, we have

$$d_{\mathcal{H}\Delta\mathcal{H}}(\sum_i \phi_i \mathbb{P}_i, \mathbb{S}) \leq \sum_i \phi_i d_{\mathcal{H}\Delta\mathcal{H}}(\mathbb{P}_i, \mathbb{S}). \tag{28}$$

Combining these two inequalities, we have

$$d_{\mathcal{H}\Delta\mathcal{H}}(\sum_i \phi_i \mathbb{P}_i, \mathbb{S}) \leq \max_{i,j} d_{\mathcal{H}\Delta\mathcal{H}}(\mathbb{P}_i, \mathbb{P}_j). \tag{29}$$

Therefore, $\mathbb{S} \in \mathcal{O}'$. Proof completed.

$\square$

# D. Methodology

## D.1. Comparisons to other methods

Vanilla Mixup [1] mixes all information including domains and classes for inputs but it only considers the outputs from classes, which leads to terrible virtual points. To make mixed inputs/features consistent with mixed outputs, FIXED generates new samples in domain-invariant features that are learned by DANN [30] or CORAL [22]. Different from MixStyle [14] and FACT [18] where more domains are generated, FIXED focuses on domain-invariant features and endeavors to make these features more diversified and discriminative.

## D.2. Novelty

FIXED is a combination of several existing algorithms, which is not our original creation. Our novelty lies in analyzing the drawbacks of existing Mixup-based approaches and proposing a simple and effective remedy to solve them. That being said, combining existing parts is easy, but understanding their limitations to know why is the main thing. Therefore, this paper can be considered as an "insight" paper rather than one claiming advance of new algorithms.

To make it simple, our novelty lies in using a simple, theoretically grounded, and effective approach to solve domain generalization problems. Now we articulate these novelties in more detail. a) In our method, Mixup is performed in specific-designed parts for domain generalization, which is totally different from Mixup [1], LISA [64], SDMix [20], and some other strategies designed for DG. Extensive experiments and ablation analysis prove the superiority of this design. b) We find the limitations of vanilla Mixup when meeting DG and introduce large margin loss to solve the issues, which is ignored in existing DG methods. And experimental results prove the effectiveness. It is not a simple combination but a specific target solution. c) Besides empirical results, we also provide novel theoretical insights to support our motivation and the proposed method.

## D.3. Limitation

Although FIXED has solved parts of the issues of vanilla Mixup, it still suffers from some other problems. For example, FIXED is not parameter-free, i.e., one should tune its hyperparameters to achieve the best performance, which is the common approach of deep learning algorithms. Then, while it can be perfectly applied to classification-based problems, more work should be done w.r.t. regression or forecasting problems since the prediction labels should be dealt with. Moreover, in Sec.6 of the main paper, we also provide some possible directions to make FIXED more complete.

## D.4. More discussion

Occasionally, more specific and diversified extracted features can bring better performance, which means that both domain and class information can work in the IID situation. For example, features learned from A+P can work well on A or P in PACS. Some existing features endeavor to combine these two parts of information for personalization [13]. However, in the OOD setting, domain-related information often interferes with models' capability on unseen targets where different distributions exist [58].

# E. Experimental Details

## E.1. Dataset details

The statistical information of each dataset is presented in Table 4 and Table 5 respectively.

UCI daily and sports dataset (DSADS) consists of 19 activities collected from 8 subjects wearing body-worn sensors on 5 body parts. USC-SIPI human activity dataset (USC-HAD) is composed of 14 subjects (7 male, 7 female, aged from 21 to 49) executing 12 activities with a sensor tied on the front right hip. UCI human activity recognition using a smartphone data set (UCI-HAR) is collected by 30 subjects performing 6 daily living activities with a waist-mounted smartphone.

Table 4: Information on visual datasets.

| Dataset | Domain Names | Domain | Class | Samples of each domain | Total Samples |
|---------|-------------|--------|-------|----------------------|---------------|
| Digits-DG | (M,MM,SVN,SYHN) | 4 | 10 | (600;600;600;600) | 2,400 |
| PACS | (A,C,P,S) | 4 | 7 | (2,048;2,344;1,670;3,929) | 9,991 |
| Office-Home | (A,C,P,R) | 4 | 65 | (2,427;4,365;4,439;4,357) | 15,588 |

Table 5: Information on HAR datasets.

| Dataset | Subjuects | Sensors | Classes | Samples |
|---------|-----------|---------|---------|---------|
| DSADS | 8 | 3 | 19 | 1,140,000 |
| USC-HAD | 14 | 2 | 12 | 5,441,000 |
| UCI-HAR | 30 | 2 | 6 | 1,310,000 |
| PAMAP | 9 | 3 | 18 | 3,850,505 |

Table 6: Information on HAR in three settings.

| Setting | Domain | Sensor | Class | Samples of each domain | Total Samples |
|---------|--------|--------|-------|----------------------|---------------|
| X-Person | 4 | 2 | 12 | (1,401,400;1,478,000;1,522,800;1,038,800) | 5,441,000 |
| X-Position | 5 | 3 | 19 | (1,140,000)*5 | 5,700,000 |
| X-Dataset | 4 | 2 | 6 | (672,000;810,550;514,950;470,850) | 2,468,350 |

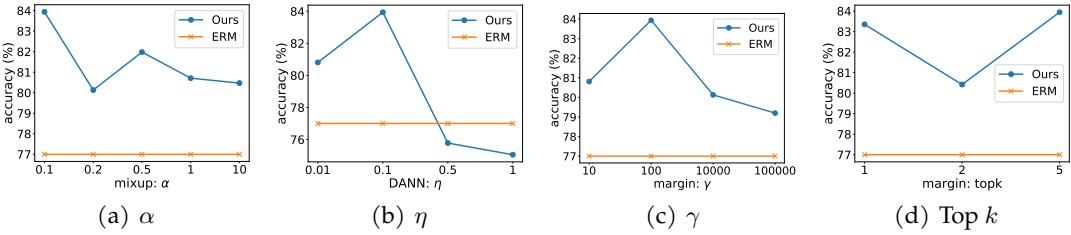

(a) $\alpha$     (b) $\eta$     (c) $\gamma$     (d) Top $k$

Figure 7: Parameter sensitivity analysis.

PAMAP2 physical activity monitoring dataset (PAMAP2) contains data of 18 different physical activities, performed by 9 subjects wearing 3 sensors.

### E.2. Details of settings on HAR

The information on three settings on HAR is shown in Table 6.

### E.3. Implementation details on HAR

We used different architectures for activity recognition. The network contains two blocks, where each has one convolution layer, one pool layer, and one batch normalization layer. A single fully-connected layer is used as the bottleneck block while another fully-connected layer serves as the classifier. In each step, each domain selects 32 samples. The maximum training epoch is set to 150. For all methods except GILE, the Adam optimizer with a learning rate $10^{-2}$ and weight decay $5 \times 10^{-4}$ is used. We tune hyperparameters for each method and select their best results to report. We report average results of three trials.

### E.4. Parameter Sensitivity Analysis

There are mainly four hyperparameters in our method: $\alpha$ for Beta distribution in Mixup, $\eta$ for the weight of adversarial learning, $\gamma$ for the required distance to boundaries in Eq. (5), and top $k$ for the aggregation class number in Eq. (5). We evaluate the parameter sensitivity of our method in Figure 7 where we change one parameter and fix the other to record the results. From these results, we can see that our method achieves better performance in a wide range, demonstrating that our method is not sensitive to hyperparameter choices. We also note that $\eta$ for DANN is a bit sensitive

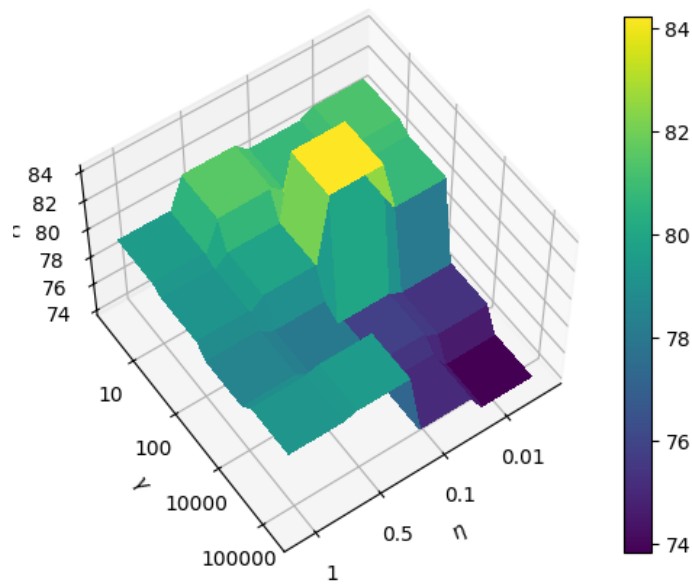

Figure 8: Parameter sensitivity analysis (2 variables).

and may need attention in real applications. We add more sensitivity analysis with two variables, i.e. $\eta$ and $\gamma$, in Figure 8. From the figure, we can see that ours performs better than ERM (77%) in a wide range. Different combinations can lead to different performances but too large $\eta$ leads to worse performance.

## E.5. Time complexity

For computational demands, FIXED consumes similar costs to ERM and Mixup. For behavior under varied conditions, we have provided some comparisons in Table 7, and we will emphasize the stable performance of FIXED compared to other methods. Table 7 shows the time costs of different methods. We can see that Mixup even costs more time FIXED since it performs mix operations in the input space.

Table 7: Time Complexity.

| Methods | ERM | Mixup | FIXED |
|---------|-----|-------|-------|
| Time (s) | 4776 | 5522 | 4940 |

