# OpenReview forum: "FIXED: Frustratingly Easy Domain Generalization with Mixup"
_CPAL.cc/2024/Conference — CPAL 2024 (Proceedings Track) Oral_

### Official Review · Reviewer_Pg71 · 2023-10-06
**Review of Submission 43**

**Rating:** 8
**Confidence:** 4

**Review:**

**Overall Evaluation**
This paper approaches domain generalization through the lens of the Mixup technique, highlighting current challenges including the entanglement of domain and class information and the potential pitfalls of introducing noisy data. Its novelty is underscored by demonstrating the adverse effects of synthetic data generation, especially in terms of domain and class entanglement, not only in conceptual illustration but also within real-world datasets.

*Pros*
1. The negative effect of entanglement on domain and class information is well proposed, and its remedy is straightforward.
2. Analytical evaluation of distribution coverage and inter-class distance further strengthens the motivation of FIXED.
3. Experiments across two modalities are well-conducted and show convincing results.


*Cons*
1. While the primary focus of this paper is on differentiating between domain and class information, offering a deeper discussion on how their relationship (i.e., entanglement) can occasionally serve beneficial roles in real-world scenarios—such as instances where a class originates from distinct domains—would enhance the paper's motivation.
2. Sensitivity analysis could be more comprehensive. The impact of adversarial learning weight and the required distance to boundaries may be intertwined. Presenting their joint effects using a 2D heatmap or 3D plot could offer deeper insights.
3. The visual presentation in this paper, including figures and tables, needs improvement. The space for Figure 4 and Table 3 should be expanded for clarity. Figure 5, in particular, appears too small and could benefit from resizing. However, these can be easily addressed in their final version.

---

### Official Review · Reviewer_nEJf · 2023-10-08
**Review for FIXED: Frustratingly Easy Domain Generalization with Mixup**

**Rating:** 5
**Confidence:** 4

**Review:**

Summary Of The Paper

This paper studies domain adaption by performing augmentations. The authors introduce a method, domain-invariant feature mixup, which is an enhancement of Mixup. This paper also introduces a margin loss to better discriminate among each class.


Main Review

- Strength
1) The theoretical validation are shown to proof its effectiveness. And the analysis on two aspects: 1) distribution coverage and 2) inter-class distance, is informative.

2) The proposed method is extensively compared against other methods on several dataset including image classification and time series, and is showing the effectiveness on moment retrieval and highlight detection tasks. And it is consistently better than other methods.

- Weakness
1) With extensive experiments and analysis, the paper has demonstrated its proposed strategy on several dataset, however, the intuition of the proposed method is not thoroughly discussed, and the novelty of this work is somewhat not very strong because the paper is adapting existing strategies (feature mixup, margin loss) together into the learning scheme.

2) In order to demonstrate the effectiveness of the proposed method, larger scale image classification dataset based on natural images, such as ImagNet must be explored, at least the author should try ImageNet-100 (https://github.com/danielchyeh/ImageNet-100-Pytorch) data, which is a subset of ImageNet. I think the proposed method may not be practical without such larger scale dataset.

Summary Of The Review

Overall, without discussing the novelty of the proposed method in details, I am not sure if the novelty that authors mention in the paper is reliable. Also, I think we need to see valid elaboration and the intuition of the proposed method, and scale up to larger dataset on image classification task. Combined with the weaknesses I mentioned above, I vote for 5. I would like to see authors response to consider raising the rating.

---

### Official Review · Reviewer_uvaF · 2023-10-15

**Rating:** 6
**Confidence:** 3

**Review:**

### Contribution:

The submission introduces FIXED, a nuanced approach aiming to refine domain generalization through a modified Mixup process. The authors articulate theoretical underpinnings for their strategy and validate their claims with experiments, suggesting broader applicability in classification tasks.

### Strengths:

1. FIXED presents a thoughtful attempt to navigate the complexities of domain-invariant representation, potentially enhancing model robustness. The approach seems promising within the scope of the presented experiments.
2. The submission delves into a detailed theoretical discourse, shedding light on the intricacies of domain generalization and the proposed method's potential advantages.
3. The empirical work is commendable, with a rigorous experimental setup and a comprehensive analysis that lends credibility to the proposed method's efficacy.

### Weaknesses:

1. The narrative detailing FIXED's operational mechanism, especially its handling of domain and class information, lacks clarity. This obscurity might hinder readers' understanding and raise questions about the method's adaptability.
2. The dense theoretical exposition in Sec. 4, while insightful, poses accessibility issues. A more approachable presentation of these complex concepts would likely benefit a wider audience.
3. The submission would gain from a more nuanced discussion on FIXED's limitations, computational demands, and its behavior under varied conditions, which remains unexplored.
4. On a minor note, the formatting issues require attention to enhance the presentation quality. Specifically, elements like Figure 4, Table 2, and Table 3 are too close to the text margins, compromising readability. Additionally, the absence of error bars in the experimental results is a concern. Incorporating error bars would significantly strengthen the solidity of the findings by providing a clearer depiction of data variability. I kindly suggest these refinements to ensure a more polished and authoritative paper.

### Broader Impact Concerns:
The submission does not directly engage with the broader implications of the proposed method. It would be beneficial for the authors to speculate on both the positive and negative ramifications of their work in real-world contexts, ensuring a holistic understanding.

### Conclusion:
This paper makes a tentative step forward in the realm of domain generalization, offering a potentially valuable method with FIXED. While the theoretical and empirical aspects are generally well-executed, there are areas where the clarity of communication and depth of analysis could be enhanced. The paper somewhat meets the conference's criteria, suggesting that, with further refinement in the areas highlighted, it could resonate with the interests of a segment of the CPAL audience. Therefore, I lean towards borderline acceptance, contingent on the authors' willingness to address the identified concerns.

---

### Official Review · Reviewer_gZbi · 2023-10-16
**solid experiments; questions on the motivation**

**Rating:** 5
**Confidence:** 3

**Review:**

Summary:

This work introduces a modified Mixup technique to address domain generation. Specifically, the mixup is performed over domain-invariant features (generated by DANN), and a large margin loss is introduced to complement the Mixup loss. Thorough experiments are conducted, together with theoretical insights.

Advantages:

• The approach is simple and easy to implement.

• Experimental evaluations are rather complete and showcase the method's efficacy.


Downsides:

• Despite the empirical improvement, the motivations are unconvincing to me. Specifically, the arguments made for Figure 1 all hinge on the fact that the toy example is very low dimensional. I am skeptical that the mixed samples will likely overlap with existing clusters. Also, I am unsure why the domain information becomes an issue -- the same argument could be made for vanilla classification tasks in one domain.

• The approach, in essence, combines several pieces of existing techniques, including domain-invariant features, manifold mixup, and margin loss, together with their hyper-parameters. So, I am unsurprised to witness the empirical gains, given the additional degrees of freedom for hyperparameter tuning.

• I would appreciate a comparison with another mix-up-based approach for distribution shift [1].


[1] https://arxiv.org/pdf/2201.00299.pdf

---

### Meta-Review · Area_Chair_ZB9b · 2023-11-13

**Recommendation:** Accept (Poster)
**Confidence:** 4

**Metareview:**

This paper proposes a modified version of Mixup for improved domain generalization. The paper provides both theoretical insights and empirical evaluation showing improvements. The reviewers generally acknowledge the quality and significance of the proposed method as well as the extensive empirical evaluation. On the negative side, 3 out of 4 reviewers point out the lack of clarity in the description of the intuition/motivations/mechanism of the proposed method. The authors should work on improving this clarity in the final version.

---

### Decision · Program_Chairs · 2023-11-19

**Decision:**

Accept (Oral)

**Comment:**

This work proposes a simple enhancement for Mixup-based domain generalization, and demonstrates its effectiveness through extensive numerical experiments, which will be a good addition to the conference. The authors should improve the writing clarity for the final paper.

The action PC chair for this paper is Yuejie Chi, who made the decision after carefully reading the paper as well as the comments by all reviewers and AC. The decision is agreed by all PC chairs.